# On-Farm Multi-Environment Evaluation of Selected Cassava (*Manihot esculenta* Crantz) Cultivars in South Africa

**DOI:** 10.3390/plants11233339

**Published:** 2022-12-01

**Authors:** Assefa B. Amelework, Michael W. Bairu, Roelene Marx, Lawrence Owoeye, Mark Laing, Sonja L. Venter

**Affiliations:** 1Agricultural Research Council, Vegetable, Industrial and Medicinal Plants, Private Bag X293, Pretoria 0001, South Africa; 2Faculty of Natural & Agricultural Sciences, School of Agricultural Sciences, Food Security and Safety Niche Area, North-West University, Private Bag X2046, Mmabatho 2735, South Africa; 3African Centre for Crop Improvement, School of Agriculture, Earth and Environmental Sciences, University of KwaZulu-Natal, Private Bag X01, Pietermaritzburg 3209, South Africa

**Keywords:** agronomic traits, genotypic performance, *Manihot esculenta*, on-farm evaluation, yield and yield related traits

## Abstract

Cassava is an important starchy root crop grown globally in tropical and subtropical regions. The ability of cassava to withstand difficult growing conditions and long-term storability underground makes it a resilient crop, contributing to food and nutrient security. This study was conducted to evaluate the performance and adaptability of exotic cassava cultivars across different environments in South Africa and to recommend genotypes for cultivation. A total of 11 cassava cultivars were evaluated at six on-farm sites, using a randomized complete block design with three replications. There were highly significant (*p* < 0.001) variations between genotypes, environments, and their interaction for all yield and yield-related traits studied. This indicates the need to test the genotypes in multiple environments before effective selection and commercialization can be undertaken. MSAF2 and UKF4 showed the overall best performances for most of the traits, whilst UKF9 (49.5%) and P1/19 (48.5%) had the highest dry matter yield. UKF4 (102.7 t ha^−1^) had the highest yield and greatest root yield stability across environments. MSAF2 did not perform consistently across environments because it was highly susceptible to cassava mosaic disease (CMD). MSAF2 could be used as a donor parent to generate novel clones with large numbers of marketable roots, and high fresh root yields, if the other parent can provide effective resistance to CMD. Based on genotype and environmental mean, Mabuyeni (KwaZulu-Natal), Mandlakazi (Limpopo), and Shatale (Mpumalanga) were found to be better environments for cassava cultivation and testing. This study is a pioneer in cassava research using multiple environments in South Africa. It provides baseline information on the performance of currently available cassava clones, their adaptation to multiple sites, the identification of suitable test sites, and information on current genetic resources for a future breeding program.

## 1. Introduction

South Africa is considered to be a food self-sufficient country in Africa. However, 20% of its rural, resource-poor households routinely experience nutritional food insecurity and poverty [1]. Furthermore, South Africa is a water-scarce country and only 12% of its land is suitable for agriculture [2]. About 1.3 million hectares of its crops are under irrigation, which consume more than 60% of the available water [3]. For the agricultural sector to play an important role in the economy, it needs to be transformed from relatively few dominant crops produced by commercial farmers into a more diverse, climate-resilient production system that cater to both commercial and small-scale farmers. Broadening the food base through the introduction, selection, and breeding for more climate-resilient crops will contribute to food and nutritional security. Cassava, being a tropical crop adaptable to diverse environments and climatic conditions, could provide South Africa with a novel food and industrial crop that could be grown widely.

Cassava is a source of dietary energy to over 800 million people in tropical and subtropical areas of the world [4], and is also used for industrial starch. It is grown primarily for its enlarged storage roots and can be harvested from 8–24 months after planting [5]. More than 40% of the African population consumes cassava as a staple food, and it is the second most important crop on the continent after maize. Globally, the mean fresh root yield of cassava is 12.8 t ha^−1^, with the highest and the lowest yields of 13.3 t ha^−1^ and 8.9 t ha^−1^ being observed in Asia and Africa, respectively [6].

Farmers can grow and harvest cassava on marginal soils with minimal inputs, on a sustainable basis, which is important to rural, resource-poor farmers [7]. Cassava can be grown on marginal lands, in low-fertility, acidic soils, and under variable rain-fed conditions ranging from less than 500 mm per year in the semi-arid tropics to more than 1000 mm per year in the sub-humid and humid tropics [8]. Due to cassava’s climate resilient nature, it is a critical crop for many farming communities and rural poor in Africa [9]. However, relatively little cassava is grown in South Africa, despite a growing community of both commercial and small-scale farmers who wish to grow the crop. Cassava can be converted into a variety of food products, and can be used as a livestock feed, and factory byproducts can be used as poultry feed [9] Furthermore, the natural high storability of cassava in the soil in comparison to other root crops allows farmers to access optimum market opportunities and earn the best possible market prices [5]. The high cost of fossil fuels, combined with the need to reduce greenhouse gas emissions, necessitates a search for renewable fuel sources. Cassava, which has a high percentage of easily fermentable sugars and a relatively high level of drought tolerance, can also be used as an alternative source of biofuel. It is also grown globally as a source of industrial starch.

It is estimated that, in South Africa, maize accounts for approximately 95% of the country’s starch production, 37% of the crop being used for food, 40% for feed, 18% for export, and 5% for industrial starch. Due to changing climatic conditions, and competition between industries utilizing maize products, the local producers of industrial starch do not meet local demands. Hence, South Africa is importing large quantities of starch annually [10]. Cassava starch is preferred in South Africa and fetches a higher price on the market than maize, potato, or wheat starch [11]. This study was initiated with the aims to evaluate the performance and adaptability of exotic cassava cultivars across different environments in South Africa and to recommend genotypes for cultivation.

## 2. Results

### 2.1. Pooled Analysis of Variance (ANOVA)

Error variance homogeneity test were made for all the environments. The residual error variances varied strongly across environments in all of the considered trials, except for the number of roots (NR), number of commercial roots (NCR), and root width (RWd). For those traits that failed the homogeneity test, a logarithm transformation (base 10) was conducted. The pooled ANOVA for 10 agro-morphological traits measured on the 11 cassava genotypes across six environments is presented in Table 1. The mean square (MS) values of both the genotype (G) and environment (E) and the genotype × environment interaction (G × E) effects were highly significant (*p* < 0.001) for all the traits under investigation. The most important source of variation for all the traits studied except HI was environment, which ranged from 41% for HI to 74% for RWt of the total sums of squares (TSS) (Figure 1). The interaction component was the highest for HI (32%) (Figure 1). For all the traits, the magnitude of the environmental variance (envi ronment and genotype x environment interaction) was higher than the genotypic variance, indicating that trait expression was highly influenced by environmental factors. The estimates of broad sense heritability varied from 19% for root weight to 93% for harvest index (Table 1). Relatively low broad sense heritability estimate was observed for root fresh yield (41%) while relatively high broad sense heritability was observed for root width (81%), number of commercial roots (74%), and dry matter content (72%).

### 2.2. Evaluation of Genotype and Environment Performances

The coefficient of variation (CV%) across the six testing environments ranged from 8.2 for DMC to 16.9 for NCR (Table 2). The CV% values for NR and NCR were particularly high (>15%). Wide variations in mean genotypic performance across environments were observed. Based on the mean performance of the 10 agro-morphological traits, Nseleni was a poor environment for cassava cultivation, while Mabuyeni, Mandlakazi, and Mutale were better environments. In Nseleni, most of the traits were poorly expressed in all the genotypes as compared to other environments, resulting in little variation in trait expression among the genotypes.

The tested genotypes performed better at Mutale with respect to root-related traits such as RWd, RWt, RL, FRY, and HI (Table 2). At Shatale, genotypes responded well for RL, DMC, and HI, whereas at Mandlakazi the NR, NCR, and PH were well differentiated. Genotypes performed well at Mabuyeni for biomass. The highest variations in the NR, PH, FRY, and BIO were observed in Mutale, whereas the highest variations in HI and DMC were recorded in Mabuyeni.

### 2.3. Mean Performance of the Genotypes

In this study, the mean NR and NCR for the tested cultivar were 7.79 and 5.16, respectively (Table 2). Table 3 presents the mean performance of the 11 cassava cultivars assessed in six environments. The cultivars UKF4 and MSAF2 were the best performing genotypes with regards to NR, NCR, RL, BIO, and FRY. Longer (>35 cm) and higher numbers of marketable roots (>5) were recorded from UKF4 and UKF9. Cultivar 98/0002 produced relatively heavier and wider roots, with a mean RWt and RWd of 1.43 kg and 7.32 cm, respectively, while UKF3 revealed a RWd of 6.67 cm and RWt of 1.48 kg. There were significant genotypic differences for FRY, ranging from 60.96-ton ha^−1^ for cultivar P1/19 to 108.85-ton ha^−1^ for UKF4, with an overall mean FRY of 83.23-ton ha^−1^.

With regards to plant height, UKF8 (2.10 m) and P4/10 (1.95 m) produced the tallest plants, and UKF7, P1/19, and 98/0002 produced the shortest plants. The highest aboveground biomass was recorded from UKF5, MSAF2, and UKF8. Wider and significant variation was observed in HI among genotypes. The overall HI mean value was 0.59 and significant higher values were observed for UKF3 (0.68), 98/0002 (0.61), and UKF9 (0.61). The DMC ranged between 30.6 and 49.5% and varied significantly (*p* < 0.001) among the tested cultivars, with the highest values being recorded for UKF9 (49.5%) and the lowest for UKF5 (30.6%) (Table 3).

### 2.4. Trait Association

The correlation coefficients for traits measured in six environments are presented in Table 4. Test for the magnitude and significance of relationships between all the measured traits revealed that approximately 54.5% (31 out of the possible 55), 5.5% (3 out of 55), and 1.8% (1 out of 55) of all potential correlations were statistically significant at (*p* < 0.001), (*p* < 0.05), and (*p* < 0.01), respectively. FRY was significantly and positively correlated (*p* ≤ 0.001) with BIO (r = 0.87), RWt (r = 0.83), NCR (r = 0.74), RWd (r = 0.68), RL (r = 0.62), PH (r = 0.44), NR (r = 0.35), and HI (r = 0.31). BIO also showed a significant positive (*p* ≤ 0.001) association with NCR (r = 0.67), RWd (r = 0.65), PH (r = 0.55), RL (r = 0.52), RWd (r = 0.52), and NR (r = 0.40). DMC was positively correlated with HI (r = 0.41). RL revealed a positive and significant association with all the traits.

### 2.5. Principal Component Analysis

The first three principal components (PCs) with an eigenvalue > 1.00 explained 79.25% of the total variation among the cultivars (Table 5). The relative magnitude of eigenvectors for the first PC was 50.35%, explained mostly by yield and root related traits, including NCR, RL, RWd, RWt, FRY, and BIO. The second and third PCs contributed 17.98 and 10.82% of the total variation, respectively. In PC2, the most predominant traits were DMC and HI, while in PC3, NR and DMC were the largest contributors.

## 3. Discussion

In plant breeding programs, multi-environment trials play an important role in cultivar evaluation and selection for primary production and commercialization. In multi-environment trials, it is commonly assumed that the residual error variance is homogenous across all considered environments. However, heterogeneity of error variances across environments generally exists in multi-environment trials [12]. Ignoring error variance differences across environments often limits the accuracy of genotype evaluations and the reliability of varietal selection [13]. Hence, an error variance homogeneity test is recommended if the research is conducted in more than one environment. In this study, the error variance was varied across environments, and logarithmic data transformation was therefore performed to standardize the error variance.

Assessment of variability is a prerequisite for crop improvement to assess the potential of the genotypes as a base for genetic improvement. Significant variability has been observed among cassava cultivars and selection of desirable characters will lead to progress in plant genetic improvement. The significant genotypic effect observed in this study signifies that the tested genotypes showed appreciable levels of variations across environments. The wide range of variability observed on the mean performance of the tested genotypes across environments suggested that there were differences in their adaptability to the different environments. Egesi [14] observed similar results for cassava FRY in multi-environmental trials. Importantly, for all the studied traits, the environmental mean square was higher than the genotypic MS. The highly significant environmental variation observed indicates that the environments of the six test sites differed greatly in temporal and spatial environmental conditions [15]. It would have been ideal if the results were supported with soil and weather information to further elucidate the cause of the significant variation in genotypic performance and environment variation. The results demonstrated the importance of the evaluation of genotypes in multiple environments with variable agro-ecological conditions before recommending a crop, and especially specific cultivars, for production and commercialization [16].

The broad sense heritability estimates for yield and yield component traits were a bit lower than the findings of Ntawuruhunga and Dixon [17] and Adjebeng-Danquah [18]. Heritability estimates influence the amount of genetic gain that can be achieved in selection for a trait of interest [19]. However, broad sense heritability does not necessarily give a full indication of genetic gain that can be achieved through selection since it includes both additive and non-additive components of the variation [20].

The different environments favoured the expression of traits differently, where some environments favoured certain genotypic performance over others. The results showed that Nseleni was a less favourable environment, and the genotypic performance of all cultivars was poor. This could be partly attributed to poor soil quality at Nseleni, which has a sandy soil with low levels of soil organic matter, suggesting that the level of N availability for optimal cassava production was too low [21]. It could also be attributed to the heavy infestation of various weeds observed at Nseleni. In cassava, weed species composition, level of infestation, and exposure time directly affect cassava yield. Maur [22] reported that weeds that were not removed for longer than 70 days could cause cassava yield losses of up to 51%.

The performance of cultivars in Mutale was better for most of the traits evaluated, compared to the rest of the environments because the cultivars were kept for longer than 12 months. Prior research has shown that the harvesting period has a strong positive association with dry matter accumulation and fresh root yield [23]. In this environment, large and longer roots were harvested. The size and shape of storage roots are also dependent on genotypic and environment factors [24].

To enhance cassava root yield and quality, it is important to understand storage root initiation and development [25]. This can be done by assessing root number, root width and length, and carbohydrate accumulation. Aina [26] reported that root traits such as NR, root size, and HI directly correlated with storage root yield. This was also confirmed in our PCA analysis, in which more than 59% the variation was contributed by root traits such as NCR, RL, RWd, and RWt. This signifies that when selecting for high FRY in cassava, these traits should be taken into consideration. A cassava plant can form up to 14 storage roots per plant, depending on the genotype [25]. In this study, a wide range of total NR (13.8) and NCR (7.8) was observed per plant, suggesting considerable scope for improvement by selection. It has also been reported that HI values higher than 0.5 can be achieved, and typically 6–12 storage roots are produced per plant at a planting density of 10,000 plants/ha [8].

The DMC of cassava roots is an important trait for the selection of cassava for industrial processing. In other studies, dry matter in cassava roots have ranged between 20 and 47%, with values above 30% considered to be high [27]. Mehouenou [28] reported an exceptionally high DMC of 55.2% for the cultivar Oueminnou. In this study, DMC ranged from 30.6% for UKF5 to 49.5% for UKF9, with a mean of 43.5%. The values observed in this study were a bit higher than the previous studies [27,29,30]. However, this could be attributed to differences in harvesting dates between the various trials. Cultivar UKF9 (49.5%) and P1/19 (48.5%) were the top performing genotypes in terms of DM yield. The high dry matter cultivars identified in this study could be grown as a feedstock for industrial starch. They would also serve as good parents in a breeding program to improve cassava DMC.

In cassava, plant height is one of the criteria used for selection of genotypes at the early stages of breeding [31]. PH is an important trait, because cassava plants are conventionally propagated using stem cuttings, hence tall plants are preferred. UKF8 was the tallest, at 2.07 m, followed by P4/10 and MSAF2. In the current study, multipurpose genotypes such as MSAF2 and UKF4 were identified with both high BIO and FRY, signifying the potential of these cultivars for food, feed, and biofuel feedstock. The advantage of using cassava as biofuel feedstock over many other crops is that cassava can thrive in degraded [32] and relatively low fertility soils, where the cultivation of other crops would be uneconomical [33]. Hence, cassava is being evaluated for the production of bioenergy, China being the leading producer [34].

In the present study, FRY was significantly and positively correlated with most of the root related traits, implying that selection for a given root trait will not have a detrimental effect on yield and other traits [35,36,37]. In cassava, storage root yield is the function of the number of storage roots per plant, mean fresh root weight, and dry matter content [38,39]. Tumuhimbise [40] also reported that FRY was positively and significantly correlated with RWd, HI, BIO, and NR. Storage root phenotyping for important yield and starch traits is often done very late in the breeding cycle, and may takes 12 to 18 months, depending on the genotypes and the environmental conditions. Hence, direct selection for root yield can be slow. However, the process can be accelerated by measuring positively correlated traits at the early growth stages of the crop [41]. Okogbenin [42] also suggested the possibility of using HI at seven months after planting as an indirect selection trait for FRY.

In this study, no significant trait association were observed between root DMC and NR, NCR, and FRY. A lack of association between DMC and FRY was also reported by Okogbenin [42], Ekanayake [43], and Rao [44]. Low correlation observed between some traits may also be beneficial in permitting independent manipulation of the traits [36,45]. Based on the result of the correlations in this study, root parameters had profound effects on final root yield [39].

## 4. Materials and Methods

### 4.1. Testing Environment

This experiment was conducted in three provinces that represents the tropical and subtropical agro-ecological zones in South Africa: KwaZulu-Natal, Mpumalanga, and Limpopo. The data presented in this report was collected from six environments, namely Nseleni, Mabuyeni, Masibekela, Shatale, Mandlakazi, and Mutale. All the trials were conducted in farmers’ fields. Detailed information on the trial site environments and GPS coordinates are presented in Figure 2.

### 4.2. Planting Material and Experimental Design

The study evaluated 11 cassava cultivars acquired from the International Institute of Tropical Agriculture (IITA), the University of KwaZulu-Natal (UKZN), and the Agricultural Research Council (ARC) (Table 6). The experiments were laid out in a randomized complete block design with three replications. Each cultivar was planted in a plot size of 25 m^2^ in five rows of 5 m long with an inter- and intra-row spacing of 1 m × 1 m. Plants were grown from disease-free, in vitro tissue cultured plantlets. The cultivars were grown under rainfed conditions, and neither fertilizers nor pesticides were applied. All other agronomic practices were followed as recommended for cassava [46].

### 4.3. Data Collected

Data was collected for 10 quantitative agronomic traits, based on cassava descriptors [47]. Data on root fresh yield, biomass, and plant height were recorded from five randomly tagged plants from the middle rows per plot. However, data on root-related traits were measured from five randomly selected roots per plant. Root number (NR) was measured by counting the number of storage roots per plant. Number of commercial roots (NCR) was measured by counting the number of roots with a length > 18 cm per plant. Root length (RL) was recorded by measuring the length (in cm) of roots from the base to the tip of the roots. Root width (RWd) was recorded by measuring the width (in cm) of roots at the middle of roots. Root weight (RWt) was measured as the weight (in gm) of roots harvested. Plant height (PH) was recorded by measuring the vertical height (in cm) of plants from the ground to the top of the canopy. Shoot biomass (BIO) was measured by weighing (in kg) the above-ground biomass of each plant. Fresh root yield (FRY) was measured by weighing (in kg) the storage roots harvested from each plant. Harvest index (HI) was measured as the ratio of the fresh root yield to above-ground shoot biomass.

Dry matter content (DMC) was recorded from five randomly selected storage roots per cultivar. The roots were thoroughly cleaned with water and dried with a paper towel. Then the roots were diced into 1 cm thick discs at 25%, 50%, and 75% of the length from the base of the roots. The freshly-cut tuber discs were further sliced into small size cubes to facilitate oven drying. Five 100 g chopped cubes were taken from each sample and were oven-dried at 105 °C for 24 h. The dried cubes were weighed to obtain the dry matter content.

DMC was measured using the following equation:$$\mathrm{Dry}\;\mathrm{matter}\;\mathrm{content}\;\left(\mathrm{DMC}\right)=\left(\frac{\mathrm{DW}}{\mathrm{FW}}\right)\times 100$$
where DW = dry weight and FW = fresh weight of the root.

### 4.4. Data Analysis

All the data generated were analysed using GenStat statistical software version 19.1 [48]. The quality of the data was inspected for data logging errors, and outliers and extreme values were removed from the analysis. Analysis of variance (ANOVA) was carried out on a plot mean basis to determine the significance of the genetic, environmental, and interaction effects of the traits measured in this study. The homogeneity of error variance of the different environments was tested using Bartlett’s homogeneity variance test. The combined ANOVA was computed over environments. The linear mixed model analysis was used, with genotype declared as fixed effect, and environment and replication as random effects using Restricted Maximum Likelihood (REML) approach in GeneStat. Principal components analysis (PCA) was conducted to identify the key factors that accounted for most of the variability in the response variables. Pearson correlation analysis was conducted to determine the association of fresh root yield to other yield attributes.
$$Y_{ijk}=\mu +R_{i\left(E\right)}+G_{j}+E_{k}+GE_{jk}+\varepsilon_{ijk}$$
where $Y_{ijk}$ = phenotypic value of the *j*th genotype harvested at the *i*th replication and at *k*th environment, *μ* = population mean, $R_{i\left(E\right)}$ = effect of the *i*th replication nested within environment, $G_{j}$ = effect of the *j*th genotype, $E_{k}$ = effect of the *k*th environment, $GE_{jk}$ = effect of the interaction between the *j*th genotype and the *k*th environment, and $\varepsilon_{ijk}$ = random error term associated with *i*th replication, *j*th genotype, and kth environment.

Heritability in the broad sense (H^2^) was estimated according to Padi [49].

## 5. Conclusions

The identification of superior genotypes and suitable testing environments based on genotypic performance is key for any breeding program. This study was conducted in six locations for one season and the effect of the seasonal variation was not considered. In most of the literature, in cassava seasonal variation was overlooked because cassava is a biannual crop that can grow from 8–24 months after planting. Ssemakula and Dixon [50] reported that location effects were more pertinent for cassava than year effects. However, Dixon and Nukenine [51] suggested that testing at 3–5 locations for 2–3 years with 3–4 replications per location is the optimum combination to get precision in cassava yield trials. It is ideal to test genotypes across locations and years to maximize genetic gain through selection.

The analysis of variance revealed the presence of highly significant variation among genotypes, environments, and their interaction. However, the magnitude of the environmental and the G × E interaction effects were significantly higher than the genetic effect for all the studied traits, indicating the need for testing cassava genotypes in multiple environments before effective selection and recommendation can be made. MSAF2 and UKF4 showed the highest mean performance for BIO, FRY, and other root traits, whilst P1/19 and UKF9 were the top in terms of DM yield. UKF4 was found to be the highest yielder, and relatively stable for root yield across environments. In contrast, MSAF2 did not perform consistently across environments because it is highly susceptible to the local strains of cassava mosaic disease (CMD). However, MSAF2 could be used as a parent to breed for novel clones that combine large number of marketable root and high fresh root yield. UKF4 can be recommended for immediate use as a food, feed, and biofuel feedstock, while P1/19 and UKF9 could be grown for industrial applications.

Understanding the relationship between environmental and demographic parameters is an important first step in predicting the quality of the testing environments. The basis for genetic improvement is to identify representative environments where the traits of interest are consistently expressed at the levels appropriate for selection [52]. Based on genotypic and environmental means, Mandlakazi and Mutale were found to be good environments for storage fresh yield, whereas Shatale and Mabuyeni were intermediate and Masibekela and Nseleni were low environments due to high CMD prevalence and weed infestation, respectively. Therefore, Mabuyeni (KwaZulu-Natal), Shatale (Mpumalanga) and Mandlakazi (Limpopo) could serve as varietal testing sites in the future cassava-breeding program. This study is a pioneering study on cassava production in South Africa, and provides baseline information on the performance of currently available cassava cultivars and the identification of suitable screening environments for future breeding and agronomic studies.

## Figures and Tables

**Figure 1 plants-11-03339-f001:**
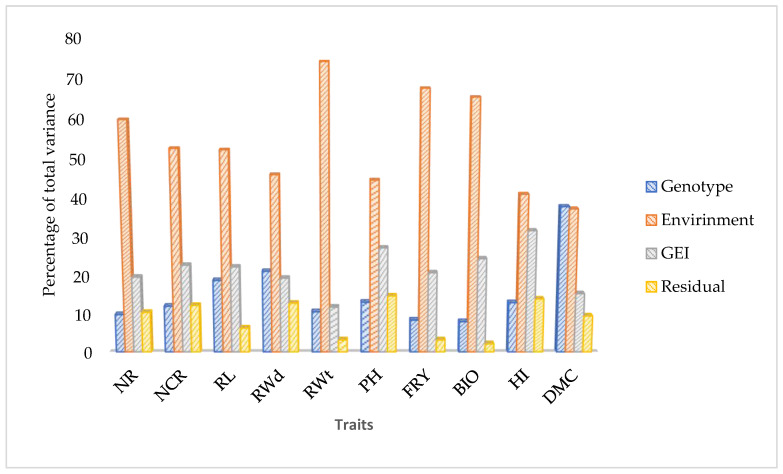
Partitioning of the total phenotypic variance into genotype, environment, GEI, and error variance. NR = Number of roots; NCR = Number of commercial roots; RWt = Root weight per plant; RL = Root length; RWd = Root width; PH = Plant height; BIO = Aboveground biomass; FRY = Fresh root yield; HI = Harvest index; DMC = Dry matter content.

**Figure 2 plants-11-03339-f002:**
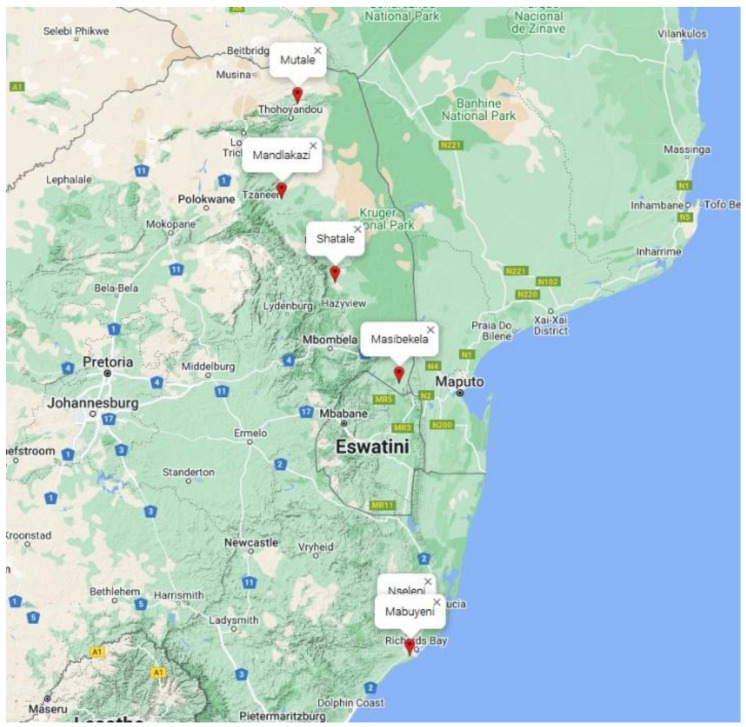
Map of South Africa showing the six trial sites.

**Table 1 plants-11-03339-t001:** Pooled ANOVA showing mean squares and percentage of total variance explained for yield and yield related traits of the cassava genotypes evaluated across six environments in South Africa.

Source	DF	NR	NCR	RL	RWd	RWt	PH	BIO	FRY	DMC	HI
Env	5	153.3 **	58.97 **	1083.60 **	35.06 **	16.66 **	1.80 **	157823 **	83075 **	877.6 **	0.33 **
Gen	10	13.4 **	6.90 **	183.18 **	6.08 **	0.62 **	0.28 **	7455 **	2026 **	476.3 **	0.04 **
Rep(E)	12	1.37	0.60	7.20	1.31	0.08	0.06	962	650	15.0	0.01
GEI	50	8.0 **	3.93 **	75.53 **	2.01 **	0.27 **	0.17 **	7015 **	2208 **	61.3 **	0.04 **
Error	120	1.47	0.76	7.60	0.44	0.03	0.03	193	118	12.7	0.01
H^2^		0.69	0.74	0.51	0.81	0.19	0.77	0.56	0.41	0.72	0.93

DF = Degree of freedom; Rep = Replication; Env = Environment; Gen = Genotype; GEI = Genotype x environment interaction; NR: Number of roots; NCR = Number of commercial roots; RWt = Root weight per plant; RL = Root length; RWd = Root width; PH = Plant height; BIO = Aboveground biomass; FRY = Fresh root yield; HI= Harvest index; DMC = Dry matter content, H2 = Broad sense heritability. ** Significant at *p* < 0.001.

**Table 2 plants-11-03339-t002:** Overall environment means, standard deviation and coefficient of variability of 10 agronomic traits measured on selected cassava genotypes evaluated in six environments.

Environment	KZN		Mpumalanga		Limpopo		Overall	
	Nseleni	Mabuyeni	Masibekela	Shatale	Mandlakazi	Mutale	Mean	CV (%)
NR	4.33 ± 0.8	8.00 ± 1.1	9.93 ± 0.9	7.19 ± 0.9	10.19 ± 1.3	7.09 ± 1.1	7.79 ± 1.0	15.6
NCR	2.86 ± 0.6	5.91 ± 0.9	4.53 ± 0.6	5.14 ± 0.7	6.61 ± 0.8	5.92 ± 1.0	5.16 ± 0.7	16.9
RL	23.5 ± 2.2	37.13 ± 2.6	30.05 ± 1.4	37.91 ± 2.3	33.28 ± 2.9	37.95 ± 2.4	33.3 ± 2.2	8.3
RWd	4.88 ± 0.6	5.87 ± 0.5	4.90 ± 0.6	6.14 ± 0.5	6.79 ± 0.5	7.48 ± 0.6	6.01 ± 0.6	11
RWt	0.48 ± 0.1	0.83 ± 0.1	0.72 ± 0.1	1.23 ± 0.2	1.69 ± 0.1	2.39 ± 0.1	1.22 ± 0.1	13.6
PH	1.38 ± 0.2	1.98 ± 0.2	1.76 ± 0.1	1.88 ± 0.2	2.04 ± 0.1	1.89 ± 0.2	1.82 ± 0.1	9.3
BIO	48.65 ± 8.2	177.32 ± 13.9	112.80 ± 13.4	108.31 ± 7.4	219.56 ± 10.4	198.14 ± 14.5	146.98 ± 12.1	9.5
FRY	31.14 ± 4.5	76.36 ± 5.1	55.49 ± 8.4	71.41 ± 10.3	117.06 ± 6.5	171.93 ± 22.3	87.23 ± 9.5	12.5
DMC	40.14 ± 1.6	38.61 ± 5.1	39.52 ± 2.0	51.59 ± 2.5	47.55 ± 2.0	43.65 ± 3.3	43.51 ± 2.9	8.2
HI	0.58 ± 0.1	0.47 ± 0.04	0.48 ± 0.05	0.68 ± 0.1	0.61 ± 0.04	0.72 ± 0.1	0.59 ± 0.1	12.6

NR = Number of roots; NCR = Number of commercial roots; RWt = Root weight per plant; RL = Root length; RWd = Root width; PH = Plant height; BIO = Above-ground biomass; FRY = Fresh root yield; HI = Harvest index; DMC = Dry matter content.

**Table 3 plants-11-03339-t003:** Mean separation analysis of 11 quantitative agronomic traits assessed in six environments in South Africa.

Geno	NR	NCR	RL	RWd	RWt	PH	BIO	FRY	HI	DMC
98/0002	6.50 ^f^	4.59 ^ef^	32.15 ^bc^	7.32 ^a^	1.43 ^a^	1.74 ^def^	149.1 ^cd^	86.77 ^cde^	0.61 ^b^	45.1 ^b^
98/0505	7.82 ^cd^	5.18 ^bcd^	31.83 ^bc^	6.21 ^c^	1.12 ^de^	1.73 ^def^	145.3 ^de^	85.59 ^de^	0.60 ^bc^	45.3 ^b^
MSAF 2	8.78 ^ab^	5.59 ^bc^	32.25 ^bc^	5.57 ^def^	1.19 ^cd^	1.89 ^bc^	160.0 ^b^	100.85 ^a^	0.58 ^bc^	41.4 ^d^
P1/19	7.97 ^cd^	4.19 ^f^	29.45 ^d^	5.33 ^f^	0.86 ^d^	1.72 ^ef^	102.8 ^g^	60.96 ^f^	0.56 ^c^	48.5 ^a^
P4/10	7.55 ^cde^	5.04 ^a^	32.55 ^b^	5.97 ^cd^	1.24 ^bc^	1.95 ^b^	150.6 ^cd^	95.02 ^ab^	0.60 ^bc^	45.7 ^b^
UKF3	7.35 ^de^	4.82 ^dc^	32.60 ^b^	6.67 ^b^	1.48 ^a^	1.72 ^ef^	135.4 ^f^	91.38 ^bcd^	0.68 ^a^	43.8 ^bc^
UKF4	9.52 ^a^	6.50 ^a^	39.50 ^a^	6.19 ^c^	1.46 ^a^	1.85 ^bc^	156.6 ^bc^	96.11 ^ab^	0.63 ^b^	42.1 ^de^
UKF5	7.36 ^de^	5.22 ^bcd^	30.55 ^cd^	5.57 ^def^	1.08 ^e^	1.84 ^bcd^	181.5 ^a^	81.77 ^e^	0.49 ^d^	30.6 ^e^
UKF7	6.77 ^ef^	4.47 ^def^	33.10 ^b^	5.92 ^cd^	1.18 ^cde^	1.69 ^f^	133.6 ^f^	82.06 ^e^	0.60 ^bc^	45.5 ^b^
UKF8	7.75 ^cd^	5.23 ^bcd^	32.11 ^b^	5.48 ^ef^	1.10 ^de^	2.10 ^a^	164.5 ^b^	80.07 ^cde^	0.55 ^c^	41.2 ^d^
UKF9	8.31 ^bc^	5.68 ^b^	39.24 ^a^	5.88 ^cdef^	1.31 ^b^	1.82 ^cde^	137.3 ^ef^	92.96 ^bc^	0.61 ^b^	49.5 ^a^

NR = Number of roots per plant; NCR = Number of commercial roots (>18 cm) per plant; RWt = Root weight per plant (kg); RL = Root length (cm); RWd = Root width (cm); PH = Plant height (cm); BIO = Biomass (t/ha); FRY = Fresh root yield (t/ha); HI = Harvest index; DMC = Dry matter content; a–f indicate different significant differences.

**Table 4 plants-11-03339-t004:** Pearson correlation coefficients the 10 quantitative traits measured on 11 cultivars evaluated at six environments in South Africa.

NR	-									
NCR	0.69 ***	-								
RL	0.41 ***	0.73 ***	-							
RWd	0.12	0.52 ***	0.52 ***	-						
RWt	0.28 ***	0.62 ***	0.61 ***	0.85 ***	-					
PH	0.40 ***	0.54 ***	0.47 ***	0.19	0.28 ***	-				
BIO	0.40 ***	0.69 ***	0.49 ***	0.54 ***	0.65 ***	0.52 ***	-			
HI	0.02	0.21 *	0.38 ***	0.51 ***	0.51 ***	0.01	−0.12	-		
DMC	0.01	0.04	0.18 *	0.20 *	0.26 **	0.10	−0.18	0.43 ***	-	
FRY	0.39 ***	0.72 ***	0.61 ***	0.72 ***	0.89 ***	0.42 ***	0.84 ***	0.36 ***	0.09	-
	NR	NCR	RL	RWd	RWt	PH	BIOM	HI	DMC	FRY

NR = Number of roots per plant; NCR = Number of commercial roots (>18 cm) per plant; RWt = Root weight per plant; RL = Root length; RWd = Root width; PH = Plant height; BIO = Biomass; FRY = Fresh root yield; HI = Harvest index; DMC = Dry matter content. *** Significant at *p* < 0.001; ** significant at *p* = 0.01; * significant at *p* = 0.5.

**Table 5 plants-11-03339-t005:** Eigenvalues, variances, and loading scores of 11 quantitative traits assessed in six environments in South Africa.

Trait	PC1	PC2	PC3
Above-ground biomass (BIOM)	0.35	−0.32	−0.31
Dry matter content (DMC)	0.08	0.48	0.50
Harvest index (HI)	0.18	0.57	0.13
No. commercial roots (NCR)	0.39	−0.19	0.18
No. roots (NR)	0.24	−0.32	0.45
Plant height (PH)	0.25	−0.28	0.37
Root length (RL)	0.35	0.02	0.21
Root width (RWd)	0.34	0.27	−0.34
Root weight (RWt)	0.39	0.21	−0.23
Root fresh yield (FRY)	0.41	0.00	−0.23
Eigenvalue	5.04	1.80	1.09
Percentage variance (%)	50.35	17.98	10.92
Cumulative variance (%)	50.35	68.33	79.25

**Table 6 plants-11-03339-t006:** Descriptions of cassava genotypes tested at six different environments.

Cultivar	Species	Type	Application	Source	Remark
98/0002	*M. esculenta*	Clones	Food and industrial	IITA	
98/0505	*M. esculenta*	Clones	Food and industrial	IITA	
MSFA2	*M. esculenta*	Clones	Food	ARC	
P1/19	*M. esculenta*	Clones	Industrial	ARC	
P4/10	*M. esculenta*	Clones	Industrial	ARC	
UKF3	*M. esculenta*	Clones	Food and industrial	UKZN	
UKF4	*M. esculenta*	Clones	Food and industrial	UKZN	
UKF5	*M. esculenta*	Clones	Food and industrial	UKZN	
UKF7	*M. esculenta*	Clones	Food and industrial	UKZN	
UKF8	*M. esculenta*	Clones	Food and industrial	UKZN	
UKF9	*M. esculenta*	Clones	Food and industrial	UKZN	

## Data Availability

Not applicable.

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
