# Peer review of "On-Farm Multi-Environment Evaluation of Selected Cassava (Manihot esculenta Crantz) Cultivars in South Africa"

_plants, 2022, doi:10.3390/plants11233339_

Round 1

Author Response

I have reviewed the manuscript # plants-1966510, titled “On-farm multi-location evaluation of selected cassava (Manihot esculenta Crantz) cultivars in South Africa”. I would like to point out some suggestions on how to improve the manuscript as follows:

  1. I would like to suggest that the material and methods section should come after the introduction section before presenting the results and others. I believe the MDPI staff will take care of that format.
  • We used the format provided on the website by the journal
  1. One of the objectives of the study is to assess the selected cassava genotypes in support of establishing cassava starch industry in South Africa. I am just curious to ask why the selected cassava clones were not evaluated for starch content and quality in addition to observed traits.
  • This study was part of a big project with the objects indicated in the manuscript. We studied the performance and adaptability of exotic cassava genotypes across different environments whereby we can recommend genotypes that are suitable for each area for production. We agreed that the stated objectives are not suitable for the particular manuscript and the objectives are modified accordingly.
  1. Secondly, the study aims to generate information on the GENETICS of yield and quality-related traits across different environments. Unfortunately, none of the results presented by the authors showed useful genetic parameters such as heritability of the trait, best linear unbiased predictions (BLUPS) value of the cassava varieties used in the study, etc. The authors used a classical pooled or combined ANOVA fixed effect model which gives the proportion of total variation attributed to each factor (genotype, environment, and their interaction). Proper estimation of those genetic parameters requires carrying out analysis in a linear mixed model framework.
  • The genotypes do not share any parental relationship and heritability is the measure of how much proportion of the phenotypic variation is due to genes being transmitted from parents to offspring. We modified the objectives.
  1. The sample size of data collected in the study is not satisfactory enough because it seems to be one-year data across 6 tested environments. There is need to have multiple years of data across the tested locations for better understanding of GxE interaction due to unpredictable pattern of a year effect.
  • Cassava is a biannual crop that grows 8 to 24 months after planting; hence, the seasonal variation is confounded within the location.
  1. In materials and method section, it is suggested that instead of presenting coordinates of study locations in tabular form (Table 1), they can be showed in a form of a map with study locations displayed as dots and labelled.
  • A map of the study locations was included
  1. In material and method section, I would like to suggest you write out the statistical model used in fitting the combined ANOVA model and describe each term. Take for example, in a single location trial where 11 genotypes were evaluated in a RCBD with 3 replicates, the statistical model takes the form �!"# = � + �! + �" + �!"# �!"# = � + �! + �" + �!"# where �!"# is a phenotypic vector of the observed trait of jth genotype from kth plot of ith block or rep and etc.
  • The model equation is included in the manuscript
  1. Going by the result presented in Table 3, the statistical model used in fitting the model seems not to be good enough and can be misleading. Each of the 6 environments in the study has its own block or rep which is a physical entity unique to each. Therefore, the model should include rep nested within environment effect i.e. rep(env) and NOT to fit main effect of rep as you did. I would suggest you refit the model as �!"# = � + �! + �" +�#(") + ��!" + �!"#. As you can see in your result the degree of freedom for rep is 2 whereas it should be 12. Rep(env) df = e*(r-1)= 6 locations * (3-1)= 12 DF.
  • We have used the same design and layout for the six locations, hence is no need of nesting the replication within the location
  1. The result presented in Table 3 looks too clumsy and this can be in supplementary. However, the result can be displayed in a graphical form for a clear understanding of readers. You can present it using bar chart where you will have the traits in the x-axis and percentage of total sum of squares (variation) along the y-axis. For each trait, you will have multiple bars corresponding to the total variation captured by environment, genotype, GxE interaction, rep(env), and residual. If you wish, you may take out rep(env) as percentage of variation attributed to it may not be of interest in your final plot.
  • A figure is included that depicts the partitioning of the total phenotypic variance into its components.
  1. It is a good idea you tested for error variance homogeneity across the environments. The idea of standardized error variances through logarithm transformation is an old-fashioned approach. The model can be fitted in a linear mixed model framework with separate error variance for each environment rather than transforming the data. May be for future study, you can explore software like SAS and ASREML-R which are more suitable for fitting such a model. You may as well explore further to know if Genstat can do the same.
  • Thank you for the advice and well noted
  1. In line 96, I think the expression in parenthesis should be (location x year). This is because the term environment is the interaction of location and year. Therefore, the environmental  variance includes variation attributed to location, year, and location-by-year interaction.
  • Comment accepted and corrected accordingly
  1. In line 101, it should be G x E = Genotype by environment interaction since your table showed G x E and NOT G x L
  • Comment accepted and corrected
  1. In line 150 where you presented the results of correlation coefficient based on the approach you used on line 322, it would be better to be specific on the kind of correlation coefficients and correlation analysis method you used. As you are aware there are different kind of correlation analysis such as Pearson, Spearman correlation, Kendall rank correlation etc. I know you used Pearson but for the sake of other readers, try to be specific.
  • Comment accepted and corrected
  1. In line 311 where you gave the formula to calculate dry matter content as DMC= DW/FW*100. The readers who are not familiar with cassava may like to know what DW and FW stand for. The abbreviations were not defined. I guess DW= dry weight and FW=  fresh weight.
  • Comment accepted and corrected
  1. In line 327 locations and their interactions may be location x year, location x genotype, or genotype x location x year. You must be consistent as you used “Env” in ANOVA table and not location. In your ANOVA table you presented ENV which I guessed denotes location x year interaction. It should probably be “The analysis of variance revealed the presence of highly significant among genotypes, environments and their interaction.”
  • Comment accepted and corrected

In summary, I recommend that the manuscript may be reconsidered suitable for publication

with major revisions. The key issue is that the authors do not have multiple years of data to

justify their findings.

Reviewer 2 Report

This manuscript is about exploratory research into the possibilities of growing cassava crops in various soil and climate conditions in South Africa, an excellent idea to make better use of soil types and cultivation conditions that do not allow for more sensitive crops. Just like in other (sub-)tropical countries, cassava is a prime candidate for such iniatives to improve the local economic situation in rural areas.

Six regions were selected in three different climate zones. More details about the respective soil and cultivation conditions are scattered through the text. I would suggest to make these more easily accesible by summarizing them in a table or perhaps add these to Table 1. A sufficient number of different crop types and regions appear to have been selected to allow for the statistical analysis of the growth-harvested results reported here. Apart from using cassava starch/products for food or cattle feed purposes, also the potential for application of cassava (starch) in industry is discussed and plants are selected for the needs of industrial application. This makes the research as such even more complete and valubale. Altogether, this manuscript appears to give valubale insights towards farmers and local policy makers, who consider to start cultivation of cassava on soil that are not all that suitable for more sensitive crops. Perhaps after more detailed study and field tests, as indicated in the conclusions section, it may still be a challenge to communicate the results and recommendations towards these usually non-scientists. Is it an idea to follow up this work with a 'public summary' towards farmers and policy makers?

The English is of good quality, the text are well readible. The research plan is clearly set up and the work appears to be well performed, also the conclusions are logical. When the relatively minor remarks and questions are answerred or processed into a few  revisions, I think this paper can be accepted for publication.

OTHER GENERAL REMARKS

I find it strange and unusual to put the section/paragraph on Materials-Methods etc AFTER the results. I have seen that in some other papers as well recently. Is this a new trend that is initiated by certain or an appointment between editors?

Please do not use that many abbreviations and genotype codes in the ABSTRACT! I am not a plant scientist, but I tend to think that for many readers, terms like 'MSAF2 and UKF4' do not ring bells.

ONE MORE SPEFIFIC QUESTION

Could issues the relatively poor growing conditions in the Nseleni area not be improved by the application of limited amounts of nitrogen-rich, (organic) fertilizers? Or would that be uneconomic with respect to the financial yield of cassava products?

Author Response

Altogether, this manuscript appears to give valubale insights towards farmers and local policy makers, who consider to start cultivation of cassava on soil that are not all that suitable for more sensitive crops. Perhaps after more detailed study and field tests, as indicated in the conclusions section, it may still be a challenge to communicate the results and recommendations towards these usually non-scientists. Is it an idea to follow up this work with a 'public summary' towards farmers and policy makers?

  • Thanks for the valuable comment, we will write a public article to communicate the results to farmers, development agents and policymakers

I find it strange and unusual to put the section/paragraph on Materials-Methods etc AFTER the results. I have seen that in some other papers as well recently. Is this a new trend that is initiated by certain or an appointment between editors?

  • We used the format provided on the website by the journal

Please do not use that many abbreviations and genotype codes in the ABSTRACT! I am not a plant scientist, but I tend to think that for many readers, terms like 'MSAF2 and UKF4' do not ring bells.

  • 'MSAF2 and UKF4' are not abbreviations, but the names of the cultivars

Could issues the relatively poor growing conditions in the Nseleni area not be improved by the application of limited amounts of nitrogen-rich, (organic) fertilizers? Or would that be uneconomic with respect to the financial yield of cassava products?

  • This project is initiated in support of resource-poor farmers. Our targets are smallholder and subsistence farmers. The cultivars were tested with zero input (no fertilizer or pesticide). Supporting and strengthening the smallholder and subsistence farmers have multiple benefits towards enhancing productivity, nutrition, and resilience in South Africa.

Round 2

Reviewer 1 Report

Dear Authors,

Kindly see the attached PDF document for my response to your feedback. Your approach to the model fitting is wrong and this has to be corrected. In addition, you did not attempt to rectify most of minor errors in my first review. 

Author Response

I have reviewed the manuscript # plants-1966510, titled “On-farm multi-location evaluation of selected cassava (Manihot esculenta Crantz) cultivars in South Africa”. I would like to point out some suggestions on how to improve the manuscript as follows:

  1. I would like to suggest that the material and methods section should come after the introduction section before presenting the results and others. I believe the MDPI staff will take care of that format.
  • We used the format provided on the website by the journal
  • Thank you, yes, I see that now.
  1. One of the objectives of the study is to assess the selected cassava genotypes in support of establishing cassava starch industry in South Africa. I am just curious to ask why the selected cassava clones were not evaluated for starch content and quality in addition to observed traits.
  • This study was part of a big project with the objects indicated in the manuscript. We studied the performance and adaptability of exotic cassava genotypes across different environments whereby we can recommend genotypes that are suitable for each area for production. We agreed that the stated objectives are not suitable for the particular manuscript and the objectives are modified accordingly.
  • The objective in the abstract (line14-18) is quite different from the objective in the last paragraph of the introduction in line 80.
  • Thank you and corrected (Lines 78-80)
  1. Secondly, the study aims to generate information on the GENETICS of yield and quality related traits across different environments. Unfortunately, none of the results presented by the authors showed useful genetic parameters such as heritability of the trait, best linear unbiased predictions (BLUPS) value of the cassava varieties used in the study etc. The authors used a classical pooled or combined ANOVA fixed effect model which gives the proportion of total variation attributed to each factor (genotype, environment, and their interaction). Proper estimation of those genetic parameters requires carrying out analysis in a linear mixed model framework.
  • The genotypes do not share any parental relationship and heritability is the measure of how much proportion of the phenotypic variation is due to genes being transmitted from parents to offspring. We modified the objectives.
  • The lack of parental relationships does not prevent the estimation of broad-sense heritability, which would be a useful addition to this study.
  • Broad sense heritability (H2) estimate was calculated for each trait and included in Table 3 and lines 96-100.
  1. The sample size of data collected in the study is not satisfactory enough because it seems to be one-year data across 6 tested environments. There is need to have multiple years of data across the tested locations for better understanding of GxE interaction due to unpredictable pattern of a year effect.
  • Cassava is biannual crop that grows 8 to 24 months after planting; hence, the seasonal variation is confounded within location.
  • Cassava can be treated as an annual crop since it can be harvested within 8 to 12 months after planting. The authors cannot at this point redo their study, but should discuss the relatively small size of the dataset and the inability to estimate genotype by year interactions.
  • The required information is included in the conclusion section (Lines 211-216)
  1. In materials and method section, it is suggested that instead of presenting coordinates of study locations in tabular form (Table 1), they can be showed in a form of a map with study locations displayed as dots and labelled.
  • Map for the study location was included
  • Thank you.
  1. In material and method section, I would like to suggest you write out the statistical model used in fitting the combined ANOVA model and describe each term. Take for example, in a single location trial where 11 genotypes were evaluated in a RCBD with 3 replicates, the statistical model takes the form �!"# = � + �! + �" + �!"# �!"# = � + �! + �" + �!"# where �!"# is a phenotypic vector of the observed trait of jth genotype from kth plot of ith block or rep and etc.
  • The model equation is included in the manuscript
  • The model Section 4.4, line 39 could be improved. Rep should be fit as rep nested within environment. rep(env). That is, the impact of Rep1 in Env1 bears no more relationship in Env2 to Rep1 than to Rep2. Therefore the effect should be nested.
  • In addition, in this same section line 33 under the section data analysis you say you use a mixed model. But there is no more mention of the random effects or of the software used to fit the mixed model, which is confusing. It seems more likely that you used a fixed effect model where the error term is a random effect by default. You claimed location and rep are random effects, but you never mentioned the probability distribution assumption associated with them. For example, when we say error is a random effect, we do say error is sampled from a normal distribution with expected mean of 0 and constant variance. You mentioned location but, in your model, you use environment. You must be consistent.
  • The model is corrected as rep nested within the location (Line 345).
  • The model is fitted using a mixed linear model in which genotype is a fixed effect and environment and rep as random effects in REML using GenStat statistical software. One of the assumptions of the linear model is the independence of observation. However, in this case, genotypes tested in one environment tend to perform similarly to those in another environment. The “correlation” of observations usually comes from some shared features by the genotypes within the same environment. To avoid the multi-collinearity of observation, considering the environment as a random effect is an efficient way to improve the estimates in the linear models.
  1. Going by the result presented in Table 3, the statistical model used in fitting the model seems not to be good enough and can be misleading. Each of the 6 environments in the study has its own block or rep which is a physical entity unique to each. Therefore, the model should include rep nested within environment effect i.e. rep(env) and NOT to fit main effect of rep as you did. I would suggest you refit the model as �!"# = � + �! + �" +�#(") + ��!" + �!"#. As you can see in your result the degree of freedom for rep is 2 whereas it should be 12. Rep(env) df = e*(r-1)= 6 locations * (3-1)= 12 DF.
  • We have used the same design and layout for the six location, hence no need of nesting the replication within location
  • Nesting is definitely needed here. It doesn’t matter if you used the same design and layout for the six locations. The rep effect must be nested within the testing locations or environment.
  • The model is corrected as suggested and the table values are corrected accordingly.
  1. The result presented in Table 3 looks too clumsy and this can be in supplementary. However, the result can be displayed in a graphical form for a clear understanding of readers. You can present it using bar chart where you will have the traits in the x-axis and percentage of total sum of squares (variation) along the y-axis. For each trait, you will have multiple bars corresponding to the total variation captured by environment, genotype, GxE interaction, rep(env), and residual. If you wish, you may take out rep(env) as percentage of variation attributed to it may not be of interest in your final plot.
  • A figure is included that depicts the partitioning of the total phenotypic variance into its components.
  • The label of Y and X-axes and the legend of a figure presented in line 106 are missing. Take for example, you have to label x-axis as “trait” to let the readers know that what you have in the x-axis are traits. The Y-axis may be labelled as Percentage of total variance explained. The legend may be labelled as “effect” or “factor”
  • The graph is corrected as suggested
  1. It is a good idea you tested for error variance homogeneity across the environments. The idea of standardized error variances through logarithm transformation is an old-fashioned approach. The model can be fitted in a linear mixed model framework with separate error variance for each environment rather than transforming the data. May be for future study, you can explore software like SAS and ASREML-R which are more suitable for fitting such a model. You may as well explore further to know if Genstat can do the same.
  • Thank you for the advice and well noted
  1. In line 96, I think the expression in parenthesis should be (location x year). This is because the term environment is the interaction of location and year. Therefore, the environmental  variance includes variation attributed to location, year, and location-by-year interaction.
  • Comment accepted and corrected accordingly
  • It has not been corrected. In line 97, the expression in parenthesis should be (location x year) and NOT (location and genotype x environment)
  • In line 94 (97 in the previous version), the authors were interpreting the data in Figure 1. In this particular case, we only assess the expression of the traits across locations but not the year. What the authors try to show is that the genotypic effect is much smaller than the environment and GEI effects.
  1. In line 101, it should be G x E = Genotype by environment interaction since your table showed G x E and NOT G x L
  • Comment accepted and corrected
  • It has not been corrected. In line 102, the footnote of Table 3 GEI = genotype x environment interaction and NOT genotype x location
  • Corrected (sorry for the silly mistake. I am working on different versions (documents) of the manuscript)
  1. In line 150 where you presented the results of correlation coefficient based on the approach you used on line 322, it would be better to be specific on the kind of correlation coefficients and correlation analysis method you used. As you are aware there are different kind of correlation analysis such as Pearson, Spearman correlation, Kendall rank correlation etc. I know you used Pearson but for the sake of other readers, try to be specific.
  • Comment accepted and corrected
  • It has not been corrected. In line 36 under the section of data analysis you mentioned correlation analysis rather than Pearson correlation analysis. In Table 6 title, it should be Pearson correlation coefficient
  • Corrected (sorry for the silly mistake. I am working on different versions (documents) of the manuscript)
  1. In line 311 where you gave the formula to calculate dry matter content as DMC= DW/FW*100. The readers who are not familiar with cassava may like to know what DW and FW stand for. The abbreviations were not defined. I guess DW= dry weight and FW=  fresh weight.
  • Comment accepted and corrected
  • It has not been corrected. In line 25 under the section data collected, there is nowhere DW and FW abbreviations are defined
  • Corrected (sorry for the silly mistake. I am working on different versions (documents) of the manuscript)
  1. In line 327 locations and their interactions may be location x year, location x genotype, or genotype x location x year. You must be consistent as you used “Env” in ANOVA table and not location. In your ANOVA table you presented ENV which I guessed denotes location x year interaction. It should probably be “The analysis of variance revealed the presence of highly significant among genotypes, environments and their interaction.”
  • Comment accepted and corrected
  • It has not been corrected. In line 48 under the section conclusion data, you still have locations rather than environments. You have to be consistent since your anova table shows environment and not locations
  • Corrected (sorry for the silly mistake. I am working on different versions (documents) of the manuscript)